# A Random Error Suppression Method Based on IGWPSO-ELM for Micromachined Silicon Resonant Accelerometers

**DOI:** 10.3390/mi14020419

**Published:** 2023-02-10

**Authors:** Peng Wang, Libin Huang, Peng Wang, Liye Zhao, Xukai Ding

**Affiliations:** 1School of Instrument Science and Engineering, Southeast University, Nanjing 210096, China; 2Key Laboratory of Micro-Inertial Instruments and Advanced Navigation Technology, Ministry of Education, Nanjing 210096, China

**Keywords:** MSRA, random error, ELM, IGWPSO

## Abstract

There are various errors in practical applications of micromachined silicon resonant accelerometers (MSRA), among which the composition of random errors is complex and uncertain. In order to improve the output accuracy of MSRA, this paper proposes an MSRA random error suppression method based on an improved grey wolf and particle swarm optimized extreme learning machine (IGWPSO-ELM). A modified wavelet threshold function is firstly used to separate the white noise from the useful signal. The output frequency at the previous sampling point and the sequence value are then added to the current output frequency to form a three-dimensional input. Additional improvements are made on the particle swarm optimized extreme learning machine (PSO-ELM): the grey wolf optimization (GWO) is fused into the algorithm and the three factors (inertia, acceleration and convergence) are non-linearized to improve the convergence efficiency and accuracy of the algorithm. The model trained offline using IGWPSO-ELM is applied to predicting compensation experiments, and the results show that the method is able to reduce velocity random walk from the original 4.3618 μg/√Hz to 2.1807 μg/√Hz, bias instability from the original 2.0248 μg to 1.3815 μg, and acceleration random walk from the original 0.53429 μg·√Hz to 0.43804 μg·√Hz, effectively suppressing the random error in the MSRA output.

## 1. Introduction

MSRA is a micro electro-mechanical system (MEMS) inertial device that uses the force-frequency characteristic of the resonator to sense acceleration. Its output is a quasi-digital signal in the form of frequency which makes it easy to detect and digitally integrate and has the advantages of strong anti-interference capability, high resolution, wide dynamic range, high sensitivity, and good stability. MSRA has shown good application prospects in navigation, automation, robotics and other fields [1,2].

However, the various types of errors in the MSRA output are cumulatively amplified during the integration process, which can lead to continuous degradation of the accuracy of the inertial guidance system or control system over time. Therefore, to reduce the impact of MSRA output errors, they should be modeled and calibrated so that they can be properly compensated or filtered before being integrated into each system [3,4,5,6].

MSRA output errors are mainly divided into deterministic and random errors. Deterministic errors are caused, on the one hand, by the limitations of the micro and nano processing level of the sensitive structure, the temperature gradient phenomenon occurring in the heat of the electronic components and the assembly errors of the sensor itself; on the other hand, they are caused by external factors such as changes in the temperature of the working environment, the influence of the magnetic field at the location of the sensor and the vibration of the carrier. The deterministic errors caused by these factors can be eliminated by calibration or by relevant theoretical calculation methods. While the random errors in MSRA are caused by the combined effect of several different types of noise sources, such as mechanical noise, electronic noise, mechanical thermal noise and environmental noise [7,8,9]. Their compositions are complex and uncertain, and cannot be eliminated by external calibration, so statistical methods or model identification theory need to be used to model the random errors.

The Allan variance method is widely used to observe and quantify random errors of sensors in the time domain by plotting the double logarithmic curve of the inertial sensor output signal and has been widely used to identify random errors in inertial sensors. For MEMS accelerometers, the types of random errors separated by the Allan variance method mainly include Quantization Noise (QN), Velocity Random Walk (VRW), Bias Instability (BI), Acceleration Random Walk (ARW), and Rate Ramp (RR) [6,7,8,9,10]. QN refers to the bias that exists when discretizing an analog signal into a digital signal, where the MSRA output is a quasi-digital signal in the form of frequency without the analog-to-digital conversion process. The RR characterizes the very slow and monotonic variation in the MSRA output over a long period of time, which makes it looks like a deterministic error generally. In contrast, VRW, BI, and ARW are the most common random errors in MSRA, as shown in Figure 1. VRW is generally generated by some thermomechanical noise and fluctuates at a rate much greater than the sampling rate of the accelerometer. BI is generated by electronic devices or other components that are susceptible to random flicker and reflects the fluctuation of accelerometer bias output at low frequency. ARW is a stochastic process with uncertain sources, which may be caused by exponential correlation noise with a long correlation time [11].

Currently, there are several methods for suppressing random errors in MEMS inertial sensors, mainly including time series analysis, sensor fusion algorithms and neural network compensation [8]. The use of time series analysis modelling and combined with Kalman filtering is more widely used [12,13,14,15,16,17,18], but this method needs to meet the requirements of zero-mean, smoothness and normality, while the output of MSRA has non-stationary and non-linear characteristics, it is difficult to meet the zero-mean requirement. In addition, the random errors are generally non-stationary processes, which require first-order or multi-order differential processing of the original data, otherwise good noise reduction cannot be obtained. Sensor fusion algorithms usually combine multiple similar or dissimilar sensors to improve the reliability of the sensor output data [19,20,21,22,23], but they still need to be coupled with filtering algorithms to achieve sufficiently high accuracy, which is not applicable to the noise reduction of a single MSRA output. Neural network techniques have shown great advantages in the field of error modelling and compensation with their non-linear, adaptive and self-learning characteristics, and the results are usually better than polynomial fitting methods [24,25,26,27,28,29,30,31,32,33,34]. Feedforward neural networks (e.g., BP, RBF neural networks) are static networks and are usually used to train time-independent data. In order to handle time series, feedback can be introduced or the network can be made memorable by using delay units to store previous states (e.g., RNN, LSTM) and the network itself becomes a dynamic system. However, real-time dynamic training networks are limited by sampling time, cannot be predicted in time to compensate for high sampling rates, and have problems with gradient disappearance or explosion for longer time series in applications.

In this paper, a single hidden layer feedforward neural network ELM is used for model training. Before training, the original signal is noise-reduced using a modified wavelet threshold function to separate the white noise from the useful signal. Then, drawing on the idea of traditional time series analysis, the MSRA output frequency at the previous sampling point and the sequence value are also used as the input dimension for the neural network training, with better prediction results. Moreover, as the ELM model sets the weights and thresholds randomly, it may lead to the model not getting the optimal solution, increasing the uncertainty in the modelling process. Therefore, by improving the PSO combined with GWO, the IGWPSO algorithm is proposed and used to optimize the initial parameters of the ELM model to avoid falling into local optimal solutions and improve the model performance. The IGWPSO-ELM is finally used for offline training and the trained model is used to suppress the random errors in the MSRA output.

## 2. Random Error Modelling Based on IGWPSO-ELM

### 2.1. Preprocessing

Complex and disordered white noise in random drifting signals of inertial sensors can affect the effectiveness of neural network modelling, e.g., a noise-contaminated neural network is prone to “overlearning” [35,36]. With this in mind, it is necessary to perform noise reduction on the original signal before ELM training. The wavelet transform, with its good time-frequency localization and multi-resolution characteristics, can overcome a series of challenges in non-stationary signal processing, and wavelet threshold noise reduction can effectively remove noise while maintaining signal singularity [37,38,39].

For a set of time series data, in general, the useful signal tends to exist in the low-frequency part of the series, while the noise signal is mostly in the high-frequency part of the series. The basic principle of the wavelet threshold noise reduction method is to process the wavelet coefficients according to the different characteristics that the wavelet coefficients of signal and noise at each scale have. The wavelet coefficients smaller than the predetermined threshold are considered to be caused by noise and are directly set to zero; the wavelet coefficients larger than the predetermined threshold are considered to be mainly caused by the useful signal and are directly retained (hard threshold method) or shrunk (soft threshold method). The estimated wavelet coefficients are finally reconstructed to obtain the denoised signal.

The hard threshold method outperforms the soft threshold method when dealing with mean square error, but the hard threshold function is not continuous at the threshold and there is a risk that the signal will oscillate at this breakpoint when reconstructed. The overall continuity of the wavelet coefficients obtained by the soft threshold method is better, but when the original set wavelet threshold is large, the output threshold quantized parameters will be much different from the original decomposed wavelet parameters, and the useful information in the high-frequency part is likely to be filtered out.

Combining the advantages and disadvantages of the above two threshold functions, this paper uses an improved wavelet threshold function for noise reduction. The form of the function is as follows, ${\bar{w}}_{j,k}$ is the wavelet estimation coefficient, *w*_*j,k*_ is the wavelet decomposition coefficient, and *λ* is the threshold.
(1)$${\bar{w}}_{j,k}=\left\{\begin{array}{ll} \mu w_{j,k}+(1-\mu )\cdot \mathrm{sgn}(w_{j,k})(|w_{j,k}|-\frac{(1-\alpha )\lambda }{\;e^{k\cdot {\left(|w_{j,k}|-\lambda \right)}^{2}}}), & |w_{j,k}|\geq \lambda \\ \mathrm{sgn}(w_{j,k})\cdot {\left(\frac{w_{j,k}}{\lambda }\right)}^{3}\alpha w_{j,k}, & |w_{j,k}|<\lambda \end{array}\right.$$

Different threshold function curves are shown in Figure 2. The curve of the improved wavelet threshold function takes ${\bar{w}}_{j,k}=w_{j,k}$ as an asymptote, i.e., when the wavelet coefficients are large enough, the new threshold function approximates the hard threshold function, solving the problem of constant deviation of the soft threshold function; at the same time with the change of parameters, the threshold changes in a wide range and the whole curve is continuous at *±λ*, effectively avoiding the defect of discontinuity of the hard threshold function at the threshold point.

There are three parameters involved in Equation (1): *k*, *α* and *μ*. Figure 3 shows the comparison of the change curves of the three parameters in the improved threshold function. In Figure 3a, as *k* increases, the curve can converge more quickly to *w*_*j,k*_, and in this paper, *k* = 30. It can be seen from the Figure 3b that with the increase in *α*, the wavelet estimation coefficient of the part less than the threshold value gradually increases, and the signal in this interval is effectively extracted instead of the accompanying noise being directly removed. After several experiments to verify, the present invention sets the value of *α* to 0.5. Considering that with the increase in *μ* in Figure 3c, the discontinuity at the threshold *±λ*, the smoothness is not good and the deviation will gradually increase, and the additional oscillation may be generated at this breakpoint when the signal is reconstructed. In order to prevent the occurrence of oscillation, the value of μ should be as small as possible, and the present invention sets the value of μ to 0.01.

The coif4 wavelet basis function with three decomposition layers and the non-linear non-stationary standard test signal *Heavy Sine* were chosen to test the noise reduction capability of the three threshold functions. Figure 4 shows the noise reduction results using different wavelet threshold functions and the signal to noise ratio (SNR) is calculated. Overall, the improved thresholding function is the best, preserving the details of the signal and reproducing the standard signal to the maximum extent. At the same time, the noise at the special inflection point can be filtered. These also confirm the effectiveness of the modified wavelet threshold noise reduction method for non-stationary data.

### 2.2. ELM

The traditional feedforward neural network is mostly trained iteratively by gradient descent, which is a slow computational method and tends to converge to a local minimum, with poor fitting accuracy. ELM improves on the back-propagation algorithm in that the connection weights between the input layer and the hidden layer, and the thresholds for the neurons in the hidden layer are generated randomly and do not need to be adjusted after they are generated. Its connection weights between the hidden and output layers are not adjusted iteratively, but are determined once by solving the equations, which improves the learning efficiency and simplifies the setting of learning parameters.

Instead of using a single output value from MSRA as the input layer, this paper draws on the idea of traditional time series analysis and uses a three-dimensional input innovatively. The idea of combining the output at the previous and current moments for optimal estimation, which is commonly used in Kalman filtering, can be introduced into neural network modelling. In addition, the introduction of the sequence value of the current sampling point is necessary to improve modelling accuracy due to the integral relationship that exists at the output of various types of random errors. Thus, the input layers of the ELM model are the current sampling point MSRA output *f_n_*, the previous sampling point MSRA output *f_n_*_−1_, and the sequence value *n* of the current sampling point, respectively, and the network structure is shown in Figure 5. The sigmoid function is chosen as the activation function *g*(*x*) for the hidden layer, the connection weights *w* between the input layer and the hidden layer and the threshold *b* of the neurons in the hidden layer are generated randomly, the connection weights *β* between the hidden layer and the output layer are calculated directly using the least squares method [40,41], and the predicted output is noted as ${\hat{f}}_{n}$.
(2)$${\hat{f}}_{n}=\sum_{i=1}^{h}\beta_{i}g(w_{1i}f_{n}+w_{2i}f_{n-1}+w_{3i}n+b_{i})$$
(3)$$g(x)=\frac{1}{1+e^{-x}}$$

Twelve sets of MSRA static output data at room temperature and 0 g were selected for offline training. After using a modified wavelet threshold function for noise reduction, nine of the data sets were used as the training set and three as the test set. The same number of neurons in the hidden layer was chosen in the same modelling platform to build the BP neural network and ELM, respectively. A random selection of 30 samples from the test set is shown in Figure 6.

The mean square error (MSE) and goodness of fit R^2^ were used as an evaluation of the accuracy of the neural network modelling and were calculated as follows:(4)$$\mathrm{MSE}=\frac{1}{N}\overset{N}{\underset{i=1}{\sum }}{\left({\hat{y}}_{i}-y_{i}\right)}^{2}$$
(5)$$R^{2}=1-\frac{\sum_{i=1}^{N}{\left(y_{i}-{\hat{y}}_{i}\right)}^{2}}{\sum_{i=1}^{N}{\left(y_{i}-\bar{y}\right)}^{2}}$$
where *N* is the number of prediction points, *y_i_* is the true value, ${\hat{y}}_{i}$ is the predicting value, $\bar{y}$ is the average of all true values. The smaller the MSE and the closer the value of R^2^ is to 1, indicating that the better the regression curve fitting effect is. A comparison of the training results of the BP neural network model and the ELM model is shown in Table 1. Since no iterative computation is required, ELM is significantly faster than the traditional BP neural network in terms of training speed, and the prediction accuracy is comparable to that of the BP network.

### 2.3. IGWPSO

The *w* and *b* in the ELM model are randomly generated, which leads to different results for each training session, so it is important to find a suitable set of weights and thresholds to make the prediction accuracy of the ELM model higher. PSO is an evolutionary algorithm proposed by Kennedy and Eberhart in 1995 to emulate the process of finding food in a flock of birds [42]. PSO is simple to implement and does not require many parameters to be adjusted, and has been widely used in function optimization, neural network training and fuzzy system control [43]. The PSO algorithm has been used in articles [44,45] to enhance the prediction accuracy and stability of the ELM, but PSO-ELM has not been applied to the field of MEMS inertial sensors at present.

Each particle in the PSO algorithm has two characteristics, velocity and position, and the fitness value is used as the criterion of evaluation. Each particle converges towards the global optimum of the particle population and its own historical optimum during each iteration, constantly evolving to find a better value. Suppose a population consists of *m* particles, the particle *i* has a spatial position *P_i_*, a velocity *V_i_*, an individual extremum *P_ibest_* and a global extremum of the population *P_gbest_*. At each iteration of the optimization, the particle updates its position *P_i_* and velocity *V_i_* according to following update formulas:(6)$$V_{i}^{k+1}=\omega V_{i}^{k}+c_{1}r_{1}(P_{i\mathrm{best}}^{k}-P_{i}^{k})+c_{2}r_{2}(P_{gbest}^{k}-P_{i}^{k})$$
(7)$$P_{i}^{k+1}=P_{i}^{k}+V_{i}^{k+1}$$
where *i* = 1, 2,..., *m*; *k* is the number of current iterations; *ω* is the inertia factor; *c*_1_ and *c*_2_ are the acceleration factors; *r*_1_ and *r*_2_ are random numbers distributed between [0, 1].

GWO is a meta-heuristic algorithm proposed by Mirjalili et al. in 2014 [46], which was proposed by imitating the social hierarchy and hunting mechanism of the grey wolves. In each iteration, *α*, *β* and *δ* represent the optimal solution, the second optimal solution and the third optimal solution respectively. The hunt (optimization) is dominated by *α*, *β* and *δ*, with *ω* wolves following these three head wolves. Assume that the target position is *X_p_*, the position of the grey wolf *i* is *X_i_*, and the grey wolf surround the prey and move towards the target at the following positions:(8)$$X_{i}^{k+1}=X_{p}^{k}-A\cdot |C\cdot X_{p}^{k}-X_{i}^{k}|$$
where *A* is a uniform random number in the interval [*−a*, *a*], *a* is a convergence factor with an initial value of 2, and *C* is a random number in the interval [0, 2].

When pursuing prey, the distance between other wolves and the three head wolves is calculated for each iteration as Equation (9), thereby updating its position as Equation (10).
(9)$$\left\{\begin{array}{l} X_{i1}^{k+1}=X_{\alpha }^{k}-A_{1}\cdot |C\cdot X_{\alpha }^{k}-X_{i1}^{k}|\\ X_{i2}^{k+1}=X_{\beta }^{k}-A_{2}\cdot |C\cdot X_{\beta }^{k}-X_{i2}^{k}|\\ X_{i3}^{k+1}=X_{\delta }^{k}-A_{3}\cdot |C\cdot X_{\delta }^{k}-X_{i3}^{k}| \end{array}\right.$$
(10)$$X_{i}^{k+1}=\frac{X_{i1}^{k+1}+X_{i2}^{k+1}+X_{i3}^{k+1}}{3}$$

Individual grey wolves attack their prey in the following way: when *|A|* < 1, the pack considers the prey as the optimal solution, and the wolves start to attack the prey and move towards the optimal solution; when |*A*| ≥ 1, the wolves move away from the prey and the pack re-finds the optimal solution globally.

The single particle swarm algorithm is limited by its own search capability and has the disadvantage of easily falling into local optimum, while GWO has good global search capability [47,48,49]. Here, the GWO algorithm is incorporated into the PSO algorithm and improved to obtain the IGWPSO algorithm to improve the performance of the PSO-ELM model.

On the one hand, the convergence of each particle in PSO towards the global optimum and its own historical optimum is replaced by each grey wolf in GWO following the three head wolves, preserving the velocity characteristics of the particle motion and taking the position of the *α* wolf as the global optimum solution. By combining Equations (6) and (9), a new particle velocity update is obtained as Equation (11).
(11)$$V_{i}^{k+1}=\omega V_{i}^{k}+c_{1}r_{1}(X_{i1}^{k}-P_{i}^{k})+c_{2}r_{2}(X_{i2}^{k}-P_{i}^{k})+c_{3}r_{3}(X_{i3}^{k}-P_{i}^{k})$$

On the other hand, there should be a larger inertia factor *ω* and a smaller acceleration factor *c* in order to improve the global exploration capability of the algorithm in the early stages, and the opposite in the later stages for local exploitation capability. The original convergence factor *a* is linearly decreasing, while the algorithm search process needs to vary non-linearly to balance the global and local search capability. Non-linearizing the inertia factor *ω*, acceleration factor *c*, and convergence factor *a* as:(12)$$\left\{\begin{array}{l} \omega^{k}=\omega_{max}-\frac{\left(\omega_{max}-\omega_{min}\right)}{2}\cdot \left(1-\cos\frac{k}{k_{max}}\pi \right)\\ c_{1,2,3}^{k}=c_{min}+\frac{\left(c_{max}-c_{min}\right)}{2}\cdot \left(1-\cos\frac{k}{k_{max}}\pi \right)\\ a=1+\cos\frac{k}{k_{max}}\pi \end{array}\right.$$
where *k_max_* is the maximum number of iterations, *ω_max_* and *ω_min_* are the maximum and minimum values of the inertia factor, and *c_max_* and *c_min_* are the maximum and minimum values of the acceleration factor.

The steps for optimizing the ELM input layer weights and implied layer thresholds using the IGWPSO algorithm are shown in Figure 7. Using the root-mean-square error as the fitness function. The comparison between PSO-ELM and IGWPSO-ELM with the same training parameters is shown in Figure 8 and Table 2. In terms of the predicted results and accuracy, the improved algorithm is closer to the real value. Meanwhile, the iterative process of two algorithms in Figure 8b shows the improved algorithm is able to search near the global optimum solution as soon as possible, and the early search efficiency is higher as IGWPSO-ELM is able to search for smaller values at the fifth iteration. IGWPSO-ELM can also jump out quickly when trapped in a local optimum, showing good local exploitation capability. Compared with the existing PSO algorithm, IGWPSO can avoid premature convergence of PSO and converge to smaller fitness values, which obviously improves the efficiency and accuracy of convergence.

## 3. Experiments and Results

### 3.1. Experiments

To verify the effectiveness of IGWPSO-ELM model, the MSRA static output measured at room temperature and 0 g was predicted and compensated. The test equipment mainly included a DC power, a high-precision turntable, and a frequency measurement module, as shown in Figure 9. The DC powered MSRA was mounted on the turntable which can provide 0 g horizontally. The output frequency of the MSRA was calculated by an FPGA in the frequency measurement module and transferred to the PC. The output frequency *f*_out_ of MSRA has the following correspondence coefficient to acceleration *a_acc_*:(13)$$f_{\mathrm{out}}=f_{0}\left(\sqrt{1+Kma_{acc}\frac{0.295L^{2}}{Ehw^{3}}}-\sqrt{1-Kma_{acc}\frac{0.295L^{2}}{Ehw^{3}}}\right)$$
where *f*_0_ is the unloaded resonant frequency of the resonator, *K* is the magnification of force, *m* is the proof mass, *L* is the length of the resonant beam, *h* is the thickness of the resonant beam, *w* is the width of the resonant beam.

### 3.2. Results and Discussion

The Allan standard deviation curves are plotted in Figure 10 comparing the uncompensated data with the compensated results for each model. Three random error coefficients are recorded in Table 3, with N, B and K corresponding to the random error coefficients for VRW, BI and ARW, respectively.

Combining Figure 10a–c, it can be seen that the IGWPSO-ELM model has the best suppression of random errors, with no obvious beats in the compensated curves and the smallest data dispersion. While the PSO-ELM model is slightly less effective than the IGWPSO-ELM in suppressing random errors, but better than the ELM model without the optimization algorithm, demonstrating the advantages of the improved optimization algorithm.

Comparing Figure 10a,d indicates that the IGWPSO-ELM after the introduction of *f_n_*_−1_ can filter out the outlier points (as circled in Figure 10a) very well, which can better restore the real useful signal while removing the influence of outliers, showing a certain noise reduction filtering ability. In addition, comparing the data of the same IGWPSO-ELM with and without the dimension *n* in Table 3 shows that the K decreases after the introduction of *n*, i.e., the part of the random error that is integrated over time is reduced.

The Allan standard deviation curve for IGWPSO-ELM (three dimensions) is found to be closer to the horizontal axis from Figure 10f, and each random error coefficient in Table 3 is also the smallest. Compared with the original data, after using the three-dimensional input IGWPSO-ELM model for prediction and compensation, VRW is reduced to 1/2 of the original, BI is reduced by approximately 1/3, and ARW is also reduced by about 1/5. These results prove that the method proposed in this paper can effectively suppress the random errors in the MSRA output, improving the output stability and measurement accuracy.

## 4. Conclusions

By analyzing the mechanism of MSRA random error generation and influencing factors, this paper introduces PSO-ELM into the field of MEMS inertial sensors and improves the optimization algorithm according to the characteristics of MSRA output which is non-stationary and non-linear. Before the training of the network, a noise reduction process is performed using an improved wavelet threshold function to eliminate the influence of white noise on the modelling of the neural network. Then, drawing on the idea of time series analysis, three-dimensional inputs are used in the input layer of the ELM for the current sampling point MSRA output *f_n_*, the previous sampling point MSRA output *f_n_*_−1_, and the sequence value of the current sampling point *n*. The increased input dimension results in higher modelling accuracy and better compensation. In order to reduce the influence of uncertainty caused by the ELM model randomly setting the connection weights *w* between the input and hidden layers and the threshold *b* of the neurons in the hidden layer, the commonly used PSO algorithm is improved by: incorporating the search mechanism of GWO; adjusting the inertia factor *ω*, acceleration factor *c* and convergence factor *a*. These can enhance the ability of the algorithm to find the global optimal solution. Finally, the trained IGWPSO-ELM model is used to predicting compensate for the MSRA static output at room temperature and 0 g state, and the results show that all coefficients of output random error for MSRA are significantly reduced, verifying the effectiveness of the IGWPSO-ELM model.

## Figures and Tables

**Figure 1 micromachines-14-00419-f001:**
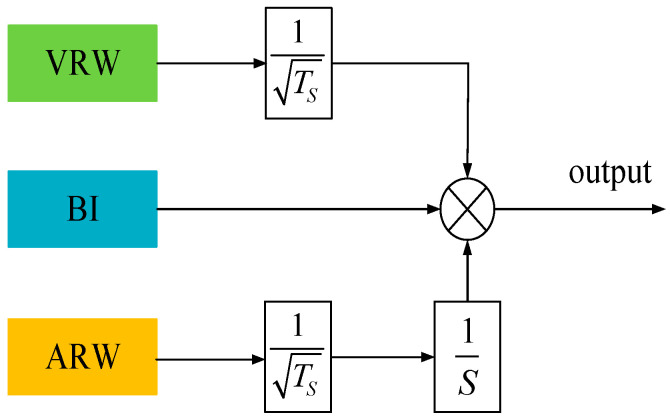
MSRA random error model (*T_S_* is the sampling time).

**Figure 2 micromachines-14-00419-f002:**
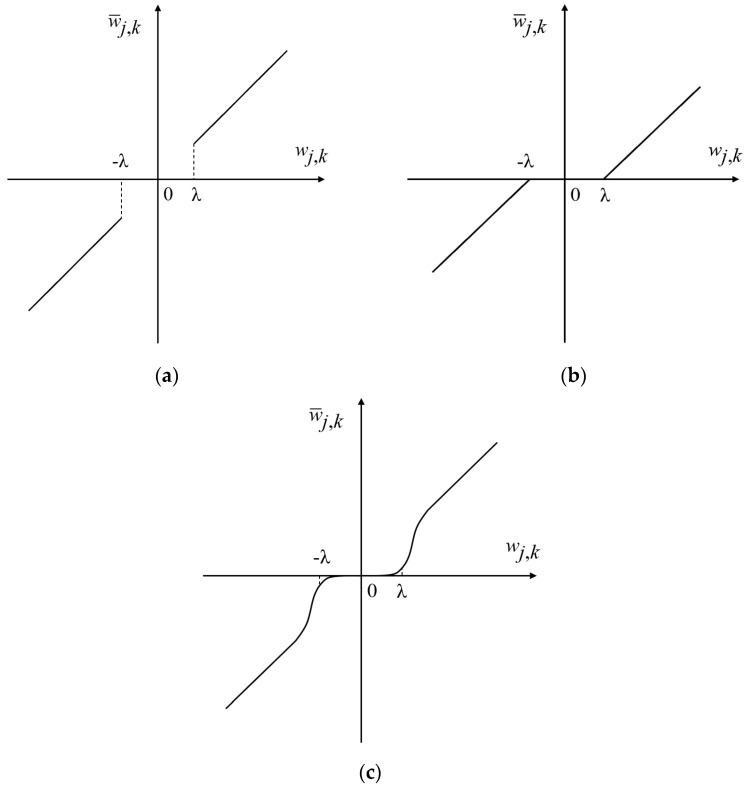
Different threshold function curves: (**a**) hard threshold function; (**b**) soft threshold function; (**c**) improved threshold function.

**Figure 3 micromachines-14-00419-f003:**
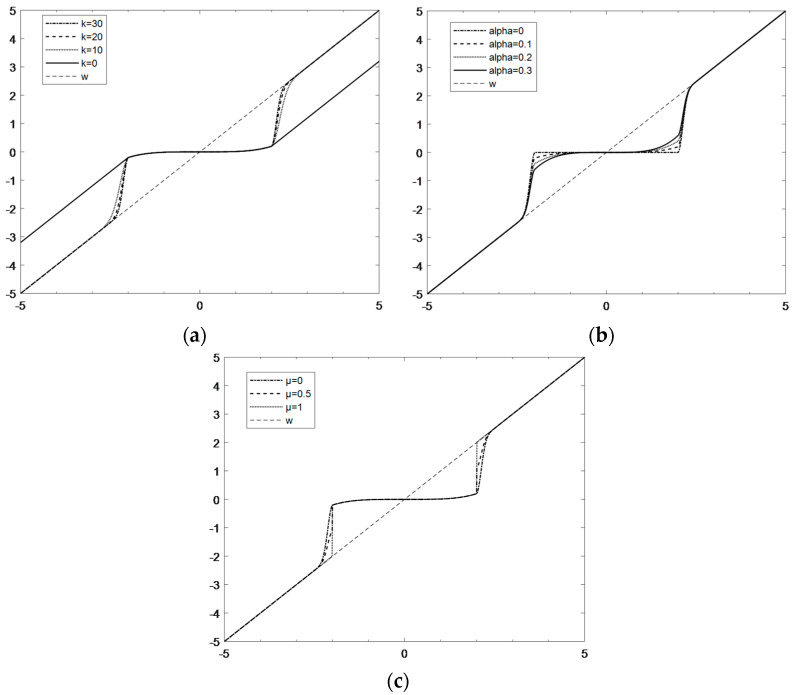
Comparison of the change curves of the three parameters in the improved threshold function: (**a**) when *α* and *μ* are fixed, *k* changes; (**b**) when *k* and *μ* are fixed, *α* changes; (**c**) when *k* and *α* are fixed, *μ* changes.

**Figure 4 micromachines-14-00419-f004:**
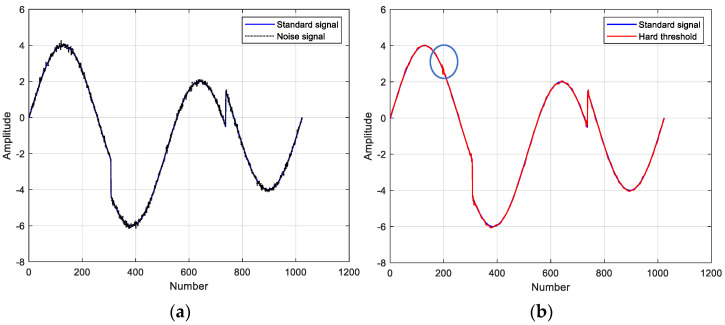
Comparison of noise reduction results with different threshold functions: (**a**) standard signal and noisy signal; (**b**) hard threshold function for noise reduction, SNR = 36.9816; (**c**) soft threshold function for noise reduction, SNR = 35.1864; (**d**) improved threshold function for noise reduction, SNR = 37.0567.

**Figure 5 micromachines-14-00419-f005:**
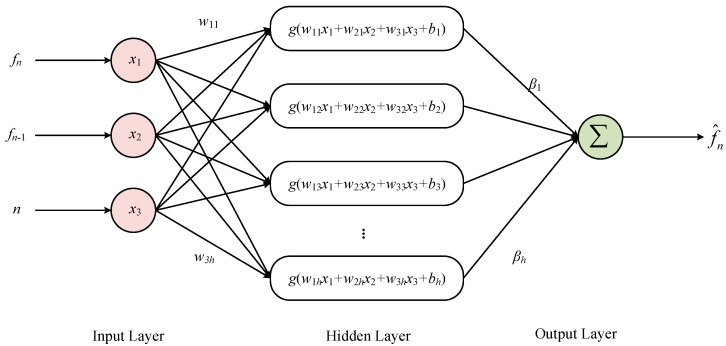
ELM network structure.

**Figure 6 micromachines-14-00419-f006:**
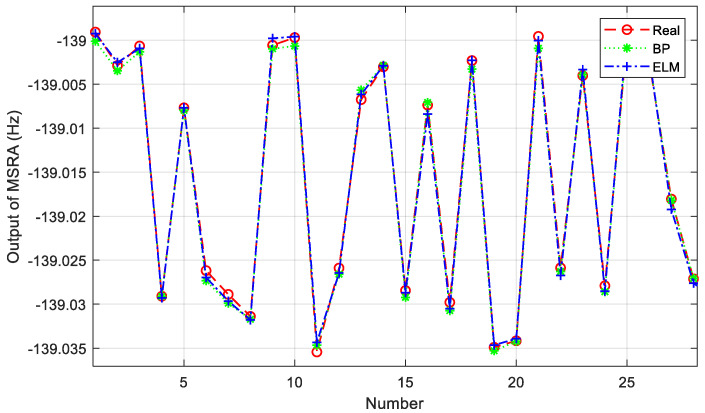
Comparison of the predicted and true values of the two models.

**Figure 7 micromachines-14-00419-f007:**
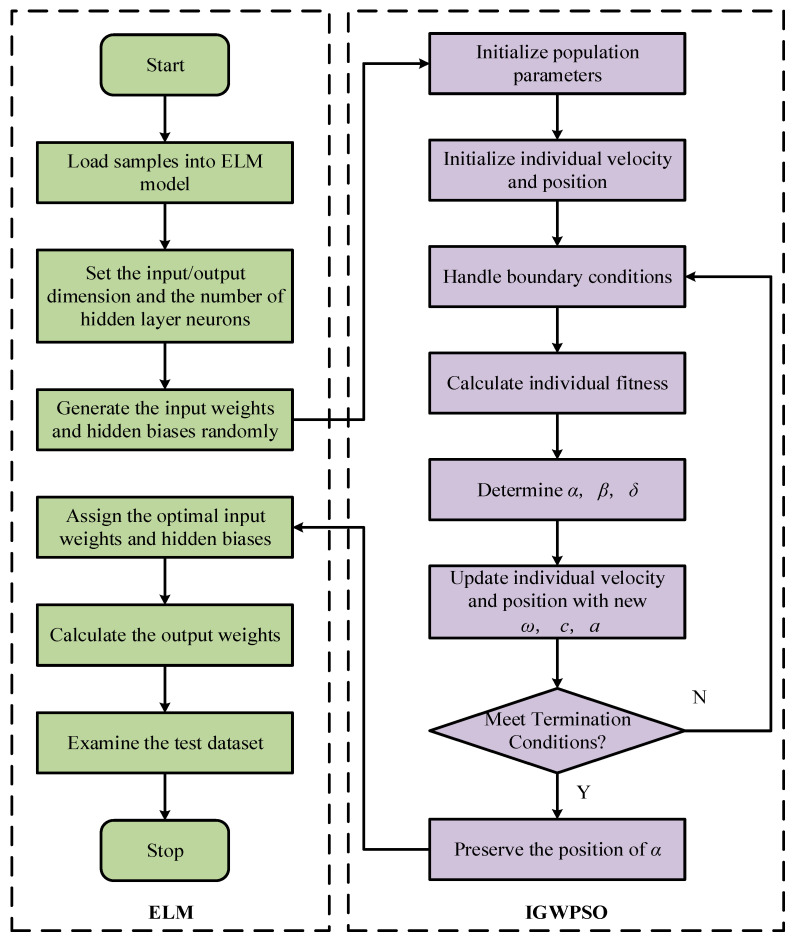
The flowchart of IGWPSO-ELM algorithm.

**Figure 8 micromachines-14-00419-f008:**
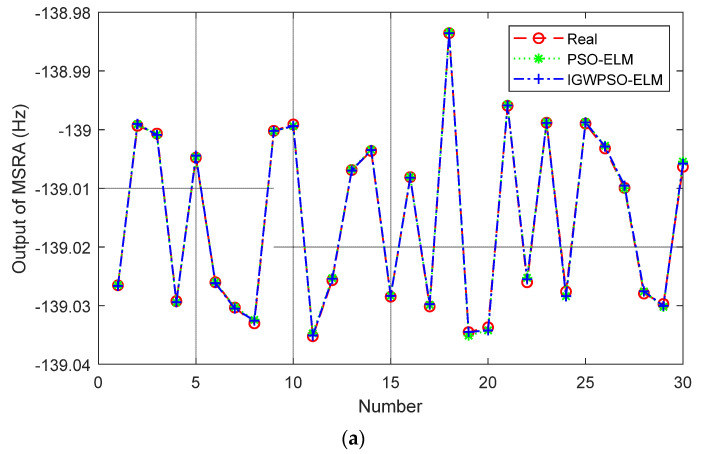
PSO-ELM vs. IGWPSO-ELM: (**a**) prediction results; (**b**) iterative process.

**Figure 9 micromachines-14-00419-f009:**
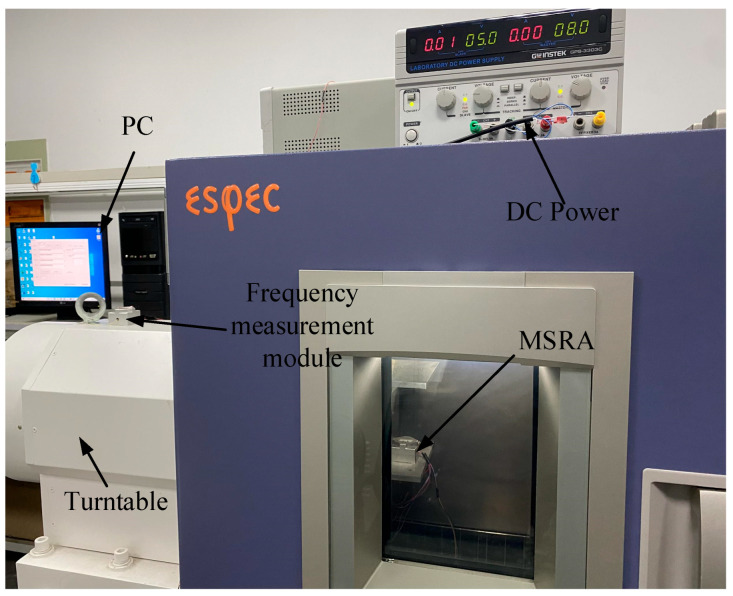
The testing environment.

**Figure 10 micromachines-14-00419-f010:**
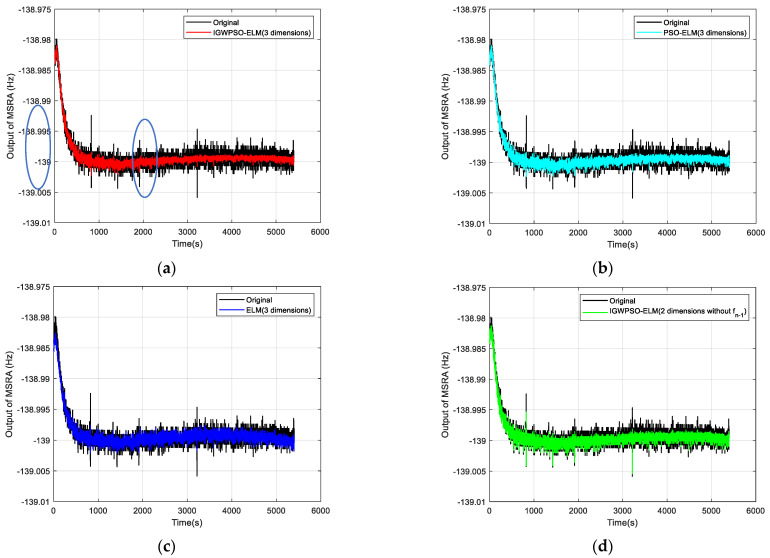
Comparison of test results: (**a**) original output with IGWPSO-ELM (three dimensions) compensated output; (**b**) original output with PSO-ELM (three dimensions) compensated output; (**c**) original output with ELM (three dimensions) compensated output; (**d**) original output with IGWPSO-ELM (two dimensions without *f_n_*_−1_); (**e**) raw output with IGWPSO-ELM (two dimensions without *n*) compensation; (**f**) Allan standard deviation curve for six data sets.

**Table 1 micromachines-14-00419-t001:** Comparison of training data between BP neural network model and ELM.

	BP	ELM
Training time (s)	1.139589	0.008554
MSE	4.3523 × 10^−7^	4.3763 × 10^−7^
R^2^	0.9982	0.99792

**Table 2 micromachines-14-00419-t002:** Comparison of the prediction accuracy between PSO-ELM and IGWPSO-ELM.

	PSO-ELM	IGWPSO-ELM
MSE	2.6929 × 10^−7^	1.2885 × 10^−7^
R^2^	0.99881	0.99974

**Table 3 micromachines-14-00419-t003:** Three random error coefficients of test data after compensated with different types of models.

	**Original output**	**IGWPSO-ELM** **(3 Dimensions)**	**PSO-ELM** **(3 Dimensions)**
*N* (μg/√Hz)	4.3618	2.1807	2.7268
*B* (μg)	2.0248	1.3815	1.6411
*K* (μg·√Hz)	0.53429	0.43804	0.47117
	**ELM** **(3 dimensions)**	**IGWPSO-ELM** **(2 dimensions without *f_n_*_−1_)**	**IGWPSO-ELM** **(2 dimensions without *n*)**
*N* (μg/√Hz)	3.4920	2.6122	2.6473
*B* (μg)	1.7988	1.4810	1.6105
*K* (μg·√Hz)	0.47583	0.43920	0.46694

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
