# Peer review of "A Random Error Suppression Method Based on IGWPSO-ELM for Micromachined Silicon Resonant Accelerometers"

_micromachines, 2023, doi:10.3390/mi14020419_

Round 1

Reviewer 1 Report

Q1In the introduction section, line 91, it has been mentioned that “However, real-time dynamic training networks are limited by sampling time, cannot be predicted in time to compensate for high sampling rates”. How to solve this issue in our work?

Q2: For the test results in Figure 9, whether the accelerometer was temperature controlled? The results were recorded since turned on or after a while? Is it possible to show the results in a clearer way?

Q3: Whether the proposed IGWPSO-ELM method can only be used for static test? How about in the practical dynamic environment?

Reviewer 2 Report

In this paper, the improved ELM algorithm is applied to the signal processing of Micromachined Silicon Resonant Accelerometers, and the suppression effect of the algorithm on random noise is confirmed by the comparison of several coefficients.

 The followings are the comments in the manuscript.

[1] In lines 56 and 61 of the article, the author describes and classifies Rate Ramp (RR) twice.

“For MEMS accelerometers, the types of random errors separated by the Allan variance method mainly include Quantization Noise (QN), Velocity Random Walk (VRW), Bias 57 Instability (BI), Acceleration Random Walk (ARW), and Rate Ramp (RR).”

“The RR characterizes the very slow and monotonic variation of the MSRA output over a long period of time, which is essentially a deterministic error

Please determine which type is correct?

 [2] In Part 2.1, the author explains the improved wavelet threshold algorithm, but for the relationship between parameters and signal-to-noise ratio, how to determine the specific values of several parameters, can the author give a more detailed explanation?

 There are many different kinds of improved threshold functions, are there any special advantages of the new threshold functions used in this article?

 The soft and hard threshold curves can be supplemented in Figure 2 to facilitate the comparison of descriptions

 [3] In section 2.2, the authors used BP to compare E LM and confirmed that ELM training time is shorter, at 2 Part 3 introduces the optimization method, but does the training time change after algorithm optimization?

 [4] The experiments given in the article are only carried out under the conditions of 0G acceleration and room temperature, can the authors verify how well the algorithm handles random noise under non-stationary variable acceleration conditions?

 [5] There is no specific explanation of Figure 7(b) in the text, please add.

[6] In Figure 9(f), when the tau value is small, the slope of several curves (IGWPSO-ELM (3 dimensions), PSO-ELM (3 dimensions), ELM (3 dimensions), IGWPSO-ELM (2 dimensions without n)) have a clear turn, why?

 Can the values of the coefficients N, B, K be added in Table 3 or Figure 9(f)?

 [7] Please add the correspondence coefficient from frequency to acceleration in the text.
